# A Hybrid of Amodiaquine and Primaquine Linked by Gold(I) Is a Multistage Antimalarial Agent Targeting Heme Detoxification and Thiol Redox Homeostasis

**DOI:** 10.3390/pharmaceutics14061251

**Published:** 2022-06-12

**Authors:** Caroline De Souza Pereira, Helenita Costa Quadros, Samuel Yaw Aboagye, Diana Fontinha, Sarah D’Alessandro, Margaret Elizabeth Byrne, Mathieu Gendrot, Isabelle Fonta, Joel Mosnier, Diogo Rodrigo M. Moreira, Nicoletta Basilico, David L. Williams, Miguel Prudêncio, Bruno Pradines, Maribel Navarro

**Affiliations:** 1Laboratório de Química Bioinorgânica e Catálise, Departamento de Química, Universidade Federal de Juiz de Fora, Juiz de Fora 36036-900, Brazil; carolsp_jf@yahoo.com.br; 2Instituto Gonçalo Moniz, Fundação Oswaldo Cruz, Salvador 40296-710, Brazil; helenita_quadros@hotmail.com (H.C.Q.); diogo.magalhaes@fiocruz.br (D.R.M.M.); 3Department of Microbial Pathogens and Immunity, Rush University Medical Center, Chicago, IL 60612, USA; samuel_y_aboagye@rush.edu (S.Y.A.); margaret_byrne@rush.edu (M.E.B.); david_williams@rush.edu (D.L.W.); 4Instituto de Medicina Molecular, Faculdade de Medicina, Universidade de Lisboa, 1649-028 Lisboa, Portugal; dfontinha@medicina.ulisboa.pt (D.F.); mprudencio@medicina.ulisboa.pt (M.P.); 5Dipartimento di Scienze Farmacologiche e Biomolecolari, Università degli Studi di Milano, 20133 Milan, Italy; sarah.dalessandro@unimi.it; 6Unité Parasitologie et Entomologie, Institut de Recherche Biomédicale des Armées, 13005 Marseille, France; ma.gendrot@laposte.net (M.G.); isabelle.fonta.09@gmail.com (I.F.); joelmosnier@orange.fr (J.M.); bruno.pradines@gmail.com (B.P.); 7Aix-Marseille Université, VITROME, 13005 Marseille, France; 8IHU Méditerranée Infection, 13005 Marseille, France; 9Centre National de Référence du Paludisme, 13005 Marseille, France; 10Dipartimento di Scienze Biomediche, Chirurgiche e Odontoiatriche, Universitá Degli Studi di Milano, 20133 Milan, Italy; nicoletta.basilico@unimi.it

**Keywords:** malaria, *Plasmodium*, antimalarial drugs, quinolines, gold, heme detoxification, redox homeostasis, hemozoin, flavoenzymes

## Abstract

Hybrid-based drugs linked through a transition metal constitute an emerging concept for *Plasmodium* intervention. To advance the drug design concept and enhance the therapeutic potential of this class of drugs, we developed a novel hybrid composed of quinolinic ligands amodiaquine (AQ) and primaquine (PQ) linked by gold(I), named [AuAQPQ]PF_6_. This compound demonstrated potent and efficacious antiplasmodial activity against multiple stages of the *Plasmodium* life cycle. The source of this activity was thoroughly investigated by comparing parasite susceptibility to the hybrid’s components, the annotation of structure–activity relationships and studies of the mechanism of action. The activity of [AuAQPQ]PF_6_ for the parasite’s asexual blood stages was influenced by the presence of AQ, while its activity against gametocytes and pre-erythrocytic parasites was influenced by both quinolinic components. Moreover, the coordination of ligands to gold(I) was found to be essential for the enhancement of potency, as suggested by the observation that a combination of quinolinic ligands does not reproduce the antimalarial potency and efficacy as observed for the metallic hybrid. Our results indicate that this gold(I) hybrid compound presents a dual mechanism of action by inhibiting the beta-hematin formation and enzymatic activity of thioredoxin reductases. Overall, our findings support the potential of transition metals as a dual chemical linker and an antiplasmodial payload for the development of hybrid-based drugs.

## 1. Introduction

Malaria is a parasitic disease caused by six *Plasmodium* species: *P. falciparum*, *P. vivax*, *P. malariae*, *P. knowlesi*, *P. ovale curtisi* and *P. ovale wallikeri*. Among these, the deadliest species are *P. falciparum* and *P. vivax*. According to the World Health Organization (WHO) in 2020, there were 241 million cases of malaria, leading to an estimated 627,000 deaths [1]. The most important strategies to prevent the disease, control infection and transmission are vaccines [2], pharmaceuticals [3] and insecticides [4]. With respect to vaccines, only one vaccine (RTS,S/AS01, Mosquirix) has been approved and recommended by the WHO for children in sub-Saharan Africa and in other areas with moderate to high *P. falciparum* transmission [5]. In the case of treatments and insecticides, the drawbacks are related to resistance and limited efficacy for certain parasite stages [3,4].

For complete elimination of malaria, new drugs acting against the hepatic, asexual blood and sexual *Plasmodium* life cycle stages are urgently needed [6]. The synthesis of hybrid-based drugs has emerged as a novel approach to obtain antiplasmodial compounds with potent activity against multiple parasite stages and strains resistant to common antimalarials [7,8,9]. The initial concept was that hybrid-based drugs could combine the antiplasmodial effects of its components (i.e., additivity or synergism) in a single molecule, facilitating the evaluation of pharmacokinetics, pharmaceutical formulation, and other aspects [7,8]. Among the concepts for designing hybrid-based drugs, there are three main possibilities: using non-cleavable chemical linkers [9,10], cleavable linkers (triggered by proteases or pH) [11,12] and transition metals that all play dual roles as cleavable chemical linkers and as antimalarials [13] (Figure 1). While all these possibilities have been investigated and have yielded promising results, the use of metals as chemical linkers offers the advantage of being pharmacologically active against parasites, while typically, organic chemical linkers are devoid of such effects.

Our strategy for metallic hybrid-based drugs was designed in line with the current combined therapy of chloroquine (CQ) and primaquine (PQ) for non-*falciparum* malaria treatment in many areas with a high incidence of *P. vivax* and *P. ovale* [14]. This combination was developed for targeting different parasite stages, with CQ killing the asexual blood stages and PQ killing hypnozoites and gametocytes. A limitation of this therapeutic combination is that while there is a consistent summation of activity for the hepatic stages of the parasite [15,16], summation of activity for the asexual blood stages is variable and depends on the parasite strain and drug ratio [17,18]. To overcome this, we previously reported the synthesis of a hybrid of PQ and CQ linked by gold(I), denoted [AuCQPQ]PF_6_ (**1**); this “three-in-one-molecule” was active against two stages of the *Plasmodium* life cycle and against three molecular targets of the parasite, confirming that it has multiple parasite stages and is a pleiotropic antimalarial agent [13]. However, hybrid (**1**) was not enough to overcome resistance to CQ when tested in vitro against CQ-resistant *P. falciparum* strains. The strategy employed here was to replace CQ by amodiaquine (AQ), leading to the [AuAQPQ]PF_6_ (**2**) compound, also referred to as hybrid (**2**) (Figure 1). 

The structure of this novel hybrid (**2**) includes components active against three parasite stages. The first component is PQ, which is active against the hepatic and sexual stages [19]. The second component is AQ, which similar to CQ, is active against the asexual blood stages through suppression of the heme detoxification process [20] and is relatively potent in vitro against hepatic stages [21]. In laboratory-cultured strains of *P. falciparum*, the CQ-resistant parasites are relatively susceptible to AQ [20]; however, in malaria patients, there is a cross-resistance of AQ to CQ and a reduced susceptibility to AQ treatment [22]. Gold(I) was chosen as the third component due to its inhibitory activity for flavoenzymes, especially the thioredoxin reductase (TrxR) of *P. falciparum* (*Pf*TrxR), which is essential for the parasite’s survival [23,24,25]. A limitation of gold(I) compounds as antimalarials is their relatively low efficacy to reduce the parasitemia in infected mice after auranofin treatment [25]. 

The hybrid compound [AuAQPQ]PF_6_ (**2**) was synthesized, and its structure was fully characterized. Multiple pharmacological studies, including a thorough investigation of its antiparasitic activity against the multiple stages of the *Plasmodium* life cycle, a comparison of parasite susceptibility to the hybrid’s components and a combination of quinoline components (AQ + PQ), and a detailed study of the mechanism of action, revealed some of the structural determinants and compelling explanations for the potency enhancement of gold(I) hybrid’s activity in comparison to its components, whether individually or in combination.

## 2. Experimental Section

### 2.1. Materials and Methods

All the gold(I) complexes were prepared under argon using common Schlenk techniques and distilled solvents prior to use. All reagents were used without prior purification. The THT-Au-Cl (THT: tetrahydrothiophene) and AuPQCl were synthesized as described in the literature [13]. All NMR experiments were recorded on a Bruker Avance III HD 500 spectrometer, 11.7 T at 298 K and using DMSO-*d*_6_ as solvent. Infrared (IR) spectra were obtained in a Bruker Alpha FT-IR Spectrometer in the region of 4000–400 cm^−1^ with a spectral resolution of 4 cm^−1^ and 64 scans. UV–Vis spectra were performed on UV-1800 Spectrophotometer Shimadzu using quartz cuvettes. Mass spectra (ESI-MS) were obtained on a 6530 Accurate-Mass Q-TOF system equipped with a dual AJS ESI source and a 1260 Infinity II high-performance liquid chromatography instrument as injection system, both from Agilent Technologies. Elemental analyses of carbon, nitrogen and oxygen were obtained in a Perkin Elmer 2400 CHNS/O Series II microanalyzer. Conductivity values were determined with a MS Tecnopon NI-CVM instrument.

### 2.2. Synthesis of [AuAQPQ]PF_6_ (2)

To a solution of AuPQCl (153 mg, 0.31 mmol) in tetrahydrofuran (10 mL), NH_4_PF_6_ (88 mg, 0.54 mmol) was added to the mixture and stirred for 1 h. The solution was then filtered through a celite pad and added dropwise to a solution of amodiaquine free base (140 mg, 0.39 mmol) in tetrahydrofuran (10 mL). After 5 h of stirring at 0 °C, the solvent was evaporated, and a brownish oil was obtained. To remove unreacted amodiaquine, acetonitrile was added leading to precipitation of final compound. The solution was separated by filtration and the solvent was evaporated to obtain a brownish oil again. Finally, the oil was washed with diethyl ether and vacuum dried, giving rise to a brownish crystalline solid. Yield: 54%; Molar conductivity in DMSO Λ_M_ DMSO = 49.30 ± 0.12 ohm^−1^ cm^2^ mol^−1^. ESI-MS (*m*/*z*): [PQ + H]^+^ = 260.1741, [AQ + H]^+^ = 356.15 [Au(AQ) + 2H]^+^ = 554.9641. IR ν 3632 (O-H); 3392, 3278 (N-H); 2972, 2942 (C-H aliphatic); 1615 (C=C); 1578 (C=N); 846 (P-F); 557 (Au-N). ^1^H NMR ((CD_3_)_2_SO) ppm 9.13 (sl, N2H); 8.39 (m, H2, H5, Hb); 8.07 (dd, J = 2 Hz, J =8 Hd); 7.87 (d, J = 2.5 Hz, H8); 7.56 (dd, J = 2 Hz, J = 9 Hz, H6); 7.42 (dd, J = 4 Hz, J = 8 Hz, Hc); 7.31 (d, J = 2.5 Hz, H3′); 7.22 (dd, J = 2.5 Hz, J = 8.5 Hz, H5′); 6.96 (d, J = 8.5 Hz, H6′); 6.61 (d, J = 5.5 Hz, H3), 6.48 (d, J = 2.5 Hz, Hg); 6.29 (d, J = 2.5 Hz, He); 6.14 (d, 9 Hz, NbH); 4.09 (s, H7′); 3.82 (s, Hl); 3.67 (m, Ha″); 2.95 (m, H8′); 2.81 (m, Hd’); 1.60 (m, Hb’, c’); 1.19 (m, H9′, a″). ^13^C{^1^H} NMR ((CD_3_)_2_SO) ppm 159.04 (Cf); 154.62 (C2′); 150.81 (C9); 149.83 (C2); 146.84 (C10); 144.66 (Ci); 144.34 (Cb); 135.09 (C7); 134.91 (Cd); 134.59 (Ch); 130.37 (Cj); 129.67 (C3′); 128.55 (C5′); 127.66 (C8); 125.60 (C4′); 125.51 (C6); 124.60 (C5); 122.21 (Cc); 118.46 (C1′); 117.28 (C4); 116.56 (C6′); 100.68 (C3); 96.34 (Ce); 91.78 (Cg); 55.05 (Cl); 50.79 (C7′); 46.88 (Ca’); 46.73 (C8′); 39.94 (Cd’); 32.81 (Cb’); 24.04 (Cc’); 20.31 (Ca’’); 8.69 (C9′). ^31^P{^1^H} NMR ((CD_3_)_2_SO) PF_6_^-^ ppm -144.21 (hep.). Anal. Calcd. (%) C_35_H_43_AuClF_6_N_6_O_2_P.3/2C_4_H_8_O: C 46.23, H 5.20, N 7.89; found: C 46.16, H 4.64, N 7.53.

### 2.3. Interaction with Ferriprotoporphyrin (Fe(III)PPIX)

The association constant (log *K*) of [AuAQPQ]PF_6_ compound with ferriprotoporphyrin IX (hemin chloride) was determined as described by Egan et al. [26] in 40% aqueous-DMSO. A hemin stock solution (pH 7.5) was prepared by adding 3.5 mg of hemin to a 10 mL of DMSO. The working solution of Fe(III)PPIX in 40% DMSO was prepared by a mixture of 140 μL of hemin stock solution, 5 mL of distilled water, 3.86 mL of DMSO and 1 mL of 0.2 M tris buffer (tris(hydroxymethyl)aminomethane). Aliquots of compounds were added to working (Fe(III)PPIX) and blank solutions in order to subtract the absorbance of the compound. The absorbance in Soret band (402 nm) was measured in the presence and absence of compound solution. The binding affinity was determined using the equation A = (A_0_ + A_∞_K[C])/(1 + K[C]) for a 1:1 complexation model using nonlinear least squares fitting, where A_0_ is absorbance of Fe(III)PPIX, A_∞_ is the absorbance of the compound-hemin adduct at saturation and *K* is the association constant. Three independent experiments were performed. 

### 2.4. Determination of β-Hematin Formation Inhibition by Infrared Spectroscopy

The conversion of hemin to β-hematin was determined by infrared (IR) spectroscopy as described by Egan et al. [26]. In a microtube, 20 mg of hemin and three equivalents of the compounds were dissolved in 3 mL of 0.1 M NaOH solution and stirred for 30 min at 60 °C. Then, 0.3 mL of 0.1 M HCl and 1.7 mL of sodium acetate buffer (10 M, pH 5) were added at the same temperature. After incubation for 120 min, the reaction mixture was cooled on ice for 10 min, and then, it was centrifuged and washed with water to remove sodium acetate salts. The solid was dried and IR spectra were obtained using attenuated total reflection (ATR) using chloroquine as a positive control.

### 2.5. β-Hematin Inhibition Activity (BHIA)

A hemin chloride (10 μL of 4 mM stock in 80% DMSO) aliquot was dispensed in a U bottom 96-well microplate, and then, 10 μL of 0.1 M NaOH (O-BHIA) or 4 mM reduced glutathione in PBS at pH 7.0 (R-BHIA) was added, and plates were sealed and incubated at 37 °C for 2 h. After this, 10 μL of the drug dissolved in DMSO was added and incubated for 2 h, followed by the addition of 180 μL of sodium acetate buffer (1.0 M, pH 5.2) and 10 μL of nonionic detergent IGEPAL-CA-630 solution in methanol (10 mg/mL). Plates were sealed and incubated at 37 °C for 18 h. Each drug was tested in a range from 1000 to 17 μM final concentration. The reaction was stopped by the addition of 50 μL of a sodium dodecyl sulfate solution (5.0% *v*/*v* in 0.1 M bicarbonate buffer at pH 9.0). Plates were centrifuged at 3700 rpm for 5 min, and 100 μL of supernatant was placed in a separate plate. Absorbance was quantified at 405 nm using a SpectraMax 190 microplate reader (Molecular Devices, Sunnyvale, CA, USA). The % inhibition was compared to untreated and unreacted controls (no acetate buffer), and IC_50_ values were estimated by using non-linear regression curve calculated in Prism version 5.01. Two independent experiments were performed, and each drug concentration was tested in quadruplicate concentrations. 

### 2.6. Interaction with Reduced Glutathione (GSH) by Fluorescence Assay

Ten microliters per well of a solution of 2.0 mM reduced glutathione (Sigma-Aldrich, St. Louis, MO, USA, dissolved in PBS pH 7.2) was distributed in a 96-well microplate containing 170 μL/well of DMSO/PBS (1:1; *v*/*v*). Stock solutions of compounds were diluted in DMSO, and 10 μL was added in quadruplicate in the respective wells to a final concentration of 0.015 to 0.5 mM. The plate was sealed and incubated at 37 °C for 2 or 24 h, an d then, the contents were transferred to an opaque plate containing 10 μL of a solution of monochlorobimane (Sigma-Aldrich) at 4.0 mM in methanol. After 1 h incubation at room temperature, fluorescence was recorded (*λ*ex = 360 nm, *λ*em = 440 nm). Blank subtraction was performed prior to monochlorobimane addition, and controls included untreated wells (no drug) and no GSH. The results were expressed as a percentage of inhibition of GSH binding to monochlorobimane in comparison to control without compound. At least two independent experiments were performed. 

### 2.7. Interaction with Reduced Glutathione by NMR

Compound (**2**) (19.7 mg, 0.021 mmol) dissolved in 300 µL of DMSO-*d_6_* and 15 mg (0.049 mmol) of reduced glutathione (GSH) dissolved in 200 µL of D_2_O were added to an NMR tube, mixed and sealed. In parallel, the same reaction mixture but containing PQ or AQ (0.021 mmol) was prepared. The ^1^H NMR spectra were recorded at different times. 

### 2.8. Inhibition of Enzymatic Activity of Recombinant Flavoenzymes

The codon-optimized sequence for *Pf*TrxR [27,28] was synthesized and subcloned into pET-15b (Genscript) and expressed and purified as for human glutathione reductase (hGR). Recombinant *Schistosoma mansoni* (*S. mansoni*) thioredoxin glutathione reductase (*Sm*TGR), human TrxR (hTrxR), and hGR proteins were expressed and purified as described [29,30,31]. TrxR and TGR enzyme inhibition assays were performed in triplicate at 25 °C in 0.1 M potassium phosphate (pH 7.4), 10 mM EDTA, 100 µM NADPH and 0.01% Tween-20. *Sm*TGR (4 nM), hTrxR (4 nM), and *Pf*TrxR (50 nM) were preincubated with the compounds for 15 min. The reaction was started with addition of an equal volume of 5,5′-dithiobis(2-nitrobenzoic acid) (6 mM) and NADPH (100 µM). The increase in A_412_ during the first 3 min was recorded. To determine inhibition of GR, hGR (120 pM) was added to an assay mixture (100 μM NADPH, 0.1 M potassium phosphate pH 6.9, 200 mM KCl, and 1 mM EDTA) with and without inhibitors, and the reaction was preincubated for 15 min. Activity was initiated with the addition of 1 mM oxidized glutathione and 100 μM NADPH, and initial rates of NADPH oxidation were monitored at 340 nm. The reactions were performed in triplicate. The IC_50_s were calculated in Prism version 5.01.

### 2.9. Cytotoxicity against Mammalian Cells

In vitro cytotoxicity was assayed against two mammalian cell lineages, J774 (murine macrophages) and HepG2 (human hepatocellular carcinoma). Lineages were maintained in RPMI-1640 (HepG2) or DMEM (J774) containing 10% fetal bovine serum and supplemented with L-glutamine, vitamins and amino acids in 75-cm^2^ flasks at 37 °C, with the medium changed twice weekly. Cell cultures from 60 to 80% confluence were trypsinized, washed in complete medium, and 4 × 10^4^ cells were plated in 100 µL per well with complete medium in 96-well flat-bottom white plates for 24 h at 37 °C prior to the addition of the compounds. Triplicate aliquots of compound and the reference drugs (stock solution in DMSO) covering 7 different concentrations at 2-fold dilutions were added to the wells, and plates were incubated for 72 h more. After incubation for 72 h at 37 °C, plates were maintained at room temperature, the culture medium was removed, and 100 µL volume of CellTiter-Glo reagent was added to each well. The bioluminescence was measured using a microplate reader Filtermax F5 Multi-Mode instrument (Molecular Devices) and Softmax software. CC_50_ data were obtained from at least two independent experiments for each cell line and analyzed using Prism version 5.01. The selectivity index (S.I.) was estimated using the IC_50_s from mammalian cells divided by the IC_50_ obtained against the asexual blood stage *P. falciparum*. 

### 2.10. P. falciparum Asexual Blood Stages

The asexual blood stages of CQ-susceptible strain (3D7) and CQ-resistant strain (W2) of *P. falciparum* were maintained in culture in RPMI-1640 supplemented with 10% human serum (Abcys S.A., Paris, France) and buffered with 25 mM 4-(2-hydroxyethyl)-1-piperazineethanesulfonic acid (HEPES) and 25 mM NaHCO_3_. Parasites were grown in A-positive human blood (Etablissement Français du Sang, Marseille, France) under controlled atmospheric conditions of 10% O_2_, 5% CO_2_ and 85% N_2_ at 37 °C with a humidity of 95%. For determination of antiparasitic activity, a volume of 25 μL/well of drug and 200 μL/well of the parasitized red blood cells at final parasitemia of 0.5% and hematocrit of 1.5% were distributed into 96-well plates. The plates were incubated for 72 h at 37 °C in controlled atmosphere. After frozen and thawing the plates, hemolyzed cultures were homogenized by vortexing the plates. The antiparasitic activity was determined using the HRP2 ELISA-based assay Malaria Ag Celisa kit (ref KM2159, Cellabs PTY LDT, Brookvale, Australia). The concentration at which the drugs were able to inhibit 50% of parasite growth (IC_50_) was calculated with the inhibitory sigmoid *E*_max_ model, with estimation of the IC_50_ through nonlinear regression using a standard function of the R software (ICEstimator version 1.2). IC_50_s are expressed as means of three to five experiments.

### 2.11. P. berghei Hepatic Stages

Human hepatoma cell line Huh-7 was cultured in 1640 RPMI medium supplemented with 10% *v*/*v* fetal bovine serum, 1% *v*/*v* nonessential amino acids, 1% *v*/*v* penicillin/streptomycin, 1% *v*/*v* glutamine, and 10 mM HEPES at pH 7 and maintained at 37 °C with 5% CO_2_. Huh-7 cells at 1.0 × 10^4^ per well were seeded in 96-well plates the day before drug treatment and infection. The culture medium was replaced by infection medium (culture medium supplemented with gentamicin (50 μg/mL) and amphotericin B (0.8 μg/mL)) containing the appropriate concentration of each compound for approximately 1 h prior to infection with sporozoites of firefly luciferase-expressing *P. berghei*, freshly obtained through the disruption of salivary glands of infected female *Anopheles stephensi* mosquitoes. An amount of the DMSO solvent equivalent to that present in the highest compound concentration was used as negative control. Sporozoite addition was followed by centrifugation at 1800× *g* for 5 min. and parasite infection load was measured 48 h after infection by using a bioluminescence assay (Biotium, Hayward, CA, USA). The effect of the compounds on the viability of Huh-7 cells was assessed by performing the AlamarBlue (Invitrogen, Waltham, MA, USA) assay according to the manufacturer’s protocol.

### 2.12. P. falciparum Gametocytes

The activity of compounds was assessed against young gametocytes (stages I/III) and against mature gametocytes (stage V). Drugs were serially diluted in a 96-well flat bottom plate (concentration range 29.0–0.22 μM) in 100 μL per well, and each drug was tested in triplicate at seven different concentrations. Methylene blue was used as positive control. A volume of 100 μL of 3D7elo1-pfs16-CBG99 gametocytes of 3D7 *P. falciparum* at 0.5–1% parasitemia and 2% hematocrit were dispensed. Plates were incubated for 72 h at 37 °C under 1% O_2_, 5% CO_2_, 94% N_2_ atmosphere. Luciferase activity was used to determine gametocytes viability. A volume of 100 µL of culture medium was removed from each well to increase hematocrit; 70 µL of resuspended culture was transferred to a black 96-well plate; 70 µL of D-luciferin (1 mM in citrate buffer 0.1 M, pH 5.5) was added. Luminescence measurements were performed after 10 min with 500 ms integration time. The IC_50_ was extrapolated from nonlinear regression analysis of the concentration–response curve. The percentage of gametocytes viability was calculated as 100 × ((OD treated sample − OD blank)/(OD untreated sample − µc-blank)) where “blank” is the sample treated with 500 nM of methylene blue, which completely kills gametocytes.

### 2.13. Mice and Parasites 

Male Swiss mice (4 to 8 weeks old) for the experimental study were obtained from the Animal Resource Facility at Instituto Gonçalo Moniz (Salvador, Brazil). Male or female Swiss mice (7 to 8 weeks old) were used for weekly passage of *P. berghei* parasites. Animals were housed under a 12 h light/dark cycle with free access to sterilized food pellets and sterilized water. These studies were approved by the Instituto Gonçalo Moniz Animal Ethics Committee (protocol 019/2021). NK-65 strain of *P. berghei* parasites expressing green fluorescent protein (GFP) was maintained by continuous weekly blood passage in mice. Mice were euthanized by tribromoethanol (300 mg/kg animal weight), blood was collected by brachial plexus, and a standard inoculum of 10^7^ parasitized erythrocytes per 200 µL was prepared by dilution in 0.9% saline and inoculated by intraperitoneal (i.p.) injection to experimental mice. Parasite enumeration in infected mice was determined in the peripheral blood collected from the tail veins. For determination of parasitemia in passage mouse, blood smear slides were mounted and stained by May–Grunwald–Giemsa. For experimental study, parasite enumeration was performed by flow cytometry using GFP signal and co-staining with a 50 nM of Mitotracker deep red FM (Life Invitrogen, Carlsbad, CA, USA) for 15 min.

### 2.14. Parasitemia Suppression in P. berghei-Infected Mice (Peters Test)

*P. berghei*-infected mice were randomly divided in *n* = 5/group. After 3 h (oral treatment by gavage) or 24 h (i.p. treatment), mice were treated once per day with 100 μL of drug or vehicle for four consecutive days. Each drug was solubilized in DMSO and diluted in a solution of kolliphor (cremophor, 4%), polysorbate 80 (5%), sorbitol (5%), glucose (5%), and tween 20 (5%) in 0.9% saline to a final concentration of 10% (*v*/*v*) of DMSO. Experimental compound [AuAQPQ]PF_6_ (**2**) was administered at 10.8 µmol/kg of animal weight (10 mg/kg), fixed-dose of drug combination (AQ + PQ) was administered at 10 µmol/kg of animal weight (using 5.0 µmol/kg each drug) or at 30 µmol/kg of animal weight (using 15 µmol/kg each drug), and untreated group received vehicle only. Twenty-four hours after last drug administration, parasitemia was determined at regular intervals for 10 days. Animal survival was monitored twice per day for at least 40 days after infection. Parasitemia reduction was determined in comparison to vehicle. Unless indicated, one single experiment was performed. 

### 2.15. Statistical Analysis

Data are presented as the median ± standard error of the mean (S.E.M.) or the mean ± standard deviation (S.D.). Statistical analysis was performed using the Prism version 5.01 software (GraphPad Software, La Jolla, CA, USA). Statistical significance was assessed by performing the one-way analysis of variance (ANOVA) followed by post-test for multiple comparisons as indicated in each figure. Differences with *p* values < 0.05 were considered significant.

## 3. Results

### 3.1. Synthesis of Metallic Hybrid and Characterization of Its Coordination Mode

The [AuAQPQ]PF_6_ (**2**) compound was synthesized in a two-step reaction as shown in Figure 2. The first step consisted of replacing the tetrahydrothiophene (THT) in the coordination sphere of the Au(THT)Cl complex with primaquine, leading to the AuPQCl intermediate. In the second step, this intermediate was dissolved in tetrahydrofuran and mixed with an excess of NH_4_PF_6_ at room temperature to remove the chloride ligand and then rapidly coordinated with amodiaquine at low temperature, which resulted in the hybrid [AuAQPQ]PF_6_. This final product was isolated as a brownish solid with a yield of 54%. [AuAQPQ]PF_6_ (**2**) was characterized by analytical and spectroscopic techniques. The IR spectrum (Appendix A) showed the characteristic and relevant bands of the quinolinic ligands. There was a slight displacement of the bands of the ligands coordinated with metal ions vis a vis the metal-free ligands. In addition to the bands of the ligands, the IR spectrum revealed the presence of bands at 557 and 846 cm^−1^, which correspond to νAu-N and νP-F, respectively, therefore confirming the formation of hybrid gold(I) linked to quinolinic ligands. The molar conductivity obtained in a DMSO solution was in a range of 1:1 electrolyte [32], and the elemental analyses confirmed the proposed molecular formula. 

The chemical assignment of proton, carbon, and phosphorus atoms was ascertained through a combination of 1D NMR spectroscopy (^1^H, ^13^C and ^31^P) and 2D NMR spectroscopy techniques, such as gradient correlation spectroscopy (^1^H-^1^H gCOSY), heteronuclear single quantum coherence (^1^H-^13^C gHSQC), and heteronuclear multiple bond coherence (^1^H-^13^C and ^1^H-^15^N) HMBC. The ^1^H NMR spectra (Appendix A) showed all the characteristic signals of quinolinic ligands and indicated that the peak integrations were consistent with the proposed molecular structure depicted in Figure 2. 

Chemical shift variation (Δδ) was used as a parameter to identify the site of coordination of the ligands to the gold(I) in comparison to the metal-free ligands. In the case of AQ, the largest Δδ values were observed for the aliphatic protons of the tertiary amine (Δδ = 0.40 ppm for H7′, Δδ = 0.44 ppm for H8′), the aromatic proton of the quinolinic ring (Δδ = 0.47 ppm for H6), and a shift in the NH proton of 0.51 ppm, all indicating that this ligand was coordinated with gold. In contrast, the main chemical shift for PQ was 0.27 ppm, which corresponds to the hydrogen Hd’ from the NH_2_ group, indicating that PQ binds gold(I) through the nitrogen in NH_2_.

In the spectra for ^13^C{^1^H}NMR (Appendix A), the signals corresponding to both ligands (AQ and PQ) were observed. The ^13^C assignment was supported by the information derived from the ^1^H-^13^C HMBC spectrum (Appendix A). The signals for Cd’ (∆δ = 2.18 ppm) of PQ, and C9′ (∆δ = 2.13 ppm), C7′ (∆δ = 3.76 ppm), C1′ (∆δ = 5.09 ppm), C8 (∆δ = 1.91 ppm), C6′ (∆δ = 2.06 ppm), C3′ (∆δ = 3.66 ppm), C7 (∆δ = 1.29 ppm), C10 (∆δ = 2.56 ppm) and C9 (1.01 ppm) of AQ were the ones most displaced in comparison with the metal-free ligands. An inspection of the ^1^H-^15^N HMBC spectra (Appendix A) further revealed that AQ coordinated with gold(I) through its quinolinic nitrogen (N1), which was inferred by the size of its shift (∆δ = −34.52) in comparison to the other nitrogen atoms from AQ (N2, ∆δ = 6.98; N3, ∆δ =12.55). 

The signals corresponding to the hydroxyl proton located at C1′ on the AQ were observed at 7.62 ppm. A plausible explanation is an interaction between the hydroxyl group and the nitrogen of the tertiary aliphatic amine, which would explain the observed displacements for the protons near the tertiary amine. Finally, the spectrum of ^31^P{^1^H}NMR (Appendix A) exhibited the septuplet at −144 ppm, which is a well-known characteristic of PF_6_ as a counter ion, and the ESI-MS spectrum of [AuAQPQ]PF_6_ (**2**) showed peaks at *m*/*z* = 260.17 and *m*/*z* = 356.15, which were attributed to the ligands PQ and AQ, respectively, in addition to a fragment detected at *m*/*z* =554.96 attributed to the cationic fragment [Au(AQ) + 2H]^+^ (Appendix A). Based on these, we propose that compound (**2**) is cationic with a linear geometry and hybridization sp. In addition to the chemical characterization, we also determined its chemical stability in DMSO solution by ^1^H and ^31^P NMR (Appendix A). No changes in the spectrum were observed for up to 25 days, indicating that [AuAQPQ]PF_6_ (**2**) is remarkably stable in solution.

### 3.2. Metallic Hybrid Is a Potent and Selective Antiparasitic for Asexual Blood Stages 

The antiplasmodial activity of compounds was first evaluated in vitro against the asexual blood stage of *P. falciparum*, and their toxicity toward mammalian cells was determined in J774 lineage (Table 1). AQ was more potent for CQ-susceptible than CQ-resistant strains of *P. falciparum*. PQ, albeit of low potency, was more active for CQ-resistant than CQ-susceptible strains. After ascertaining this, the activity of [AuAQPQ]PF_6_ (**2**) was determined, and it displayed potent activity. For the CQ-susceptible strain, it was slightly less potent than CQ and AQ. For the CQ-resistant strain, [AuAQPQ]PF_6_ (**2**) was 29-fold more potent than AQ and 154-fold more potent than CQ. 

Determination of cytotoxicity for mammalian cells revealed that [AuAQPQ]PF_6_ (**2**) has a profile that is similar to its quinoline components with a cytotoxic effect that is lower than the reference drug (doxorubicin, CC_50_ = 0.44 ± 0.31 μM for J774; CC_50_ < 0.12 μM for HepG2). Both PQ and AQ were more cytotoxic for J774 cells than cancerous HepG2 cells, while [AuAQPQ]PF_6_ (**2**) was equally cytotoxic regardless of the lineage (Table 1 and Table 2). The overall cytotoxicity of [AuAQPQ]PF_6_ (**2**) was in the micromolar range, providing a selectivity index superior to PQ and comparable to AQ. 

Given the high selectivity index of [AuAQPQ]PF_6_ (**2**) as an antiplasmodial agent for the asexual blood stages of *P. falciparum,* we tested its efficacy as a blood schizonticidal agent in *P. berghei*-infected mice using a standard 4-day regime. A dose of 10.8 μmol/kg of animal weight of [AuAQPQ]PF_6_ (**2**) was chosen based on a previous dose–response curve for hybrid (**1**) [13]. Efficacy of [AuAQPQ]PF_6_ (**2**) was evaluated both by intraperitoneal injection and orally by gavage. As shown in Figure 3, compound (**2**) given by intraperitoneal injection was efficacious in reducing parasitemia and increasing animal survival in comparison to untreated infected group. When given orally, (**2**) suppressed parasitemia, but with a cure rate of only 50% from a pool of two independent experiments. Tested in parallel at the same dose by oral administration, a fixed-dose drug combination (AQ + PQ) reduced parasitemia but did not extend the median of animal survival. This drug combination only extended the median of animal survival when given at a dose of 30 μmol/kg, although this was not enough to cure the mice. Based on this, we estimated that the [AuAQPQ]PF_6_ (**2**) is at least three-fold more efficacious as a blood schizonticidal agent in *P. berghei*-infected mice than a drug combination (AQ + PQ). 

### 3.3. Metallic Hybrid Is Active against the Sexual and Hepatic Stages 

The antiplasmodial activity of [AuAQPQ]PF_6_ (**2**) was subsequently evaluated against the hepatic stages using Huh-7 cells infected with sporozoites of *P. berghei* and against the sexual blood stage (gametocytes) of the 3D7 strain of *P. falciparum* (Table 2). AQ is relatively potent against the *Plasmodium* hepatic stage, being more potent than PQ and other quinolines; however, AQ has a low potency for the gametocytes. PQ is active against the sexual and hepatic stages, but it depends on *in vivo* metabolism [20], as it typically exhibits low in vitro potency. Of note, a representative gold(I) complex lacking quinolinic ligand [AuClPPh_3_] exhibits low activity for these parasite stages (IC_50_ of 15.2 ± 2.6 μM for gametocytes V; no inhibitory activity for the hepatic stages up to 10 μM). 

Having ascertained the antiplasmodial profile of hybrid components, [AuAQPQ]PF_6_ (**2**) was tested and found to have potent activity against both parasite stages. For the hepatic stages, [AuAQPQ]PF_6_ (**2**) at 10 μM was more potent than PQ, AQ, and their combination, and its activity was achieved with an excellent selectivity index, as it was not cytotoxic for Huh-7 cells. For the sexual stages, the activity was assessed in young (stages I to III) and in mature gametocytes (stage V). For young gametocytes, the [AuAQPQ]PF_6_ (**2**) compound displayed IC_50_ value in low nanomolar range, comparable to the reference drug dihydroartemisinin. Moreover, the activity of hybrid (**2**) against young gametocytes was similar to the activity against the asexual blood stages. For mature gametocytes, it presented potency in micromolar range, which is higher than PQ or AQ but lower than the gametocidal drug of reference methylene blue. We further compared the activity of [AuAQPQ]PF_6_ (**2**) against gametocytes with its analog [AuCQPQ]PF_6_ (**1**, IC_50_ of 0.057 ± 0.012 and 8.3 ± 0.91 μM for gametocytes I to III and V, respectively), and the results revealed that, in essence, gold(I) hybrids have a similar gametocidal activity, i.e., a potent activity against young gametocytes but limited activity against mature gametocytes. 

### 3.4. Metallic Hybrid Targets the Plasmodium Heme Detoxification

To study the potential of [AuAQPQ]PF_6_ (**2**) for suppressing *Plasmodium* heme detoxification, the association constant (log *K*) values of compounds to bind to the soluble Fe(III)-PPIX and the ability of compounds to suppress formation of the β-hematin crystals (the synthetic counterpart of hemozoin) were determined.

The binding of [AuAQPQ]PF_6_ with Fe(III)-PPIX was examined by spectroscopic titration of UV–Vis using the Soret band. In this experiment, a quenching of the Soret band is typically observed upon increasing concentration of compounds, indicating the drug:heme binding process. Based on the log *K* values tabulated in Figure 4, it was observed that the affinity of [AuAQPQ]PF_6_ (**2**) for binding Fe(III)-PPIX is similar as observed for AQ. This contrasts to a previous hybrid [AuCQPQ]PF_6_ (**1**), which had a higher affinity than its quinoline component CQ (Figure 5). We posit that a decrease in the Fe(III)-PPIX affinity by [AuAQPQ]PF_6_ is due to AQ being coordinated through its quinolinic nitrogen, which is a structural requirement for Fe(III)-PPIX binding. 

The inhibition of β-hematin formation was monitored by infrared spectroscopic analysis of the two bands at 1660 and 1207 cm^−1^ that are observed with the dimer but are absent from the monomer [26]. The β-hematin formation was inhibited by [AuAQPQ]PF_6_ as well as by the quinolinic compounds (AQ and CQ) but not by PQ (negative control) (Figure 3 and Appendix A). To determine a compound’s potency in inhibiting β-hematin formation, the β-hematin inhibitory activity (BHIA) was assayed in oxidized iron (Fe(III)PPIX, O-BHIA) and reduced iron (Fe(II)PPIX, R-BHIA) conditions [33]. [AuAQPQ]PF_6_ (**2**) displayed potent BHIA against the ferric hematin with a potency that is twice that of CQ and almost as potent as AQ, while PQ and artemisinin lacked inhibitory activity. [AuAQPQ]PF_6_ was only slightly more potent for ferrous heme than ferric hematin, while CQ was twice as potent for ferrous heme than ferric hematin. Both hybrid (**2**) and AQ were potent for R-BHIA, albeit they were less potent than artemisinin, which is a strong inhibitor in this assay. Based on this, we can infer that [AuAQPQ]PF_6_ (**2**) is an inhibitor of β-hematin crystal formation, which is consistent with its strong binding to Fe(III)-PPIX. 

### 3.5. Metallic Hybrid Inhibits Flavoenzymes Involved in the Thiol Redox Homeostasis

*Plasmodium* relies on the flavoenzymes glutathione reductase (GR) and TrxR to control its thiol redox homeostasis [34], and gold(I) compounds such as auranofin can presumably cause an imbalance of this homeostasis by inhibiting the enzymatic activity of these enzymes. To understand the effects of [AuAQPQ]PF_6_ (**2**) in this homeostasis, its inhibition of the enzymatic activity was determined against a panel of flavoenzymes: *Pf*TrxR, hTrxR, hGR, and *Sm*TGR. The results, displayed in Table 3, show that AQ did not inhibit the enzymatic activity of any flavoenzyme with appreciable potency. In contrast, [AuAQPQ]PF_6_ (**2**) inhibited hTrxR and *Sm*TGR with potency in the low micromolar range; in common, these two are selenocysteine (Sec) TrxRs. For *Pf*TrxR, which is a cysteine (Cys) TrxR, [AuAQPQ]PF_6_ (**2**) did not show appreciable inhibition. This inhibitory profile was explained when [AuAQPQ]PF_6_’s activity was compared to its analog [AuCQPQ]PF_6_ (**1**) and to the gold(I) phosphine compounds auranofin and [AuCl(PPh_3_)]. Similar to the [AuAQPQ]PF_6_ (**2**), the CQ analog [AuCQPQ]PF_6_ (**1**) had more potent inhibition of the Sec-TrxRs than the Cys-TrxR and hGR. Auranofin was found to be a pan-inhibitor for all TrxRs but not for GR, while [AuClPPh_3_] was found to be an inhibitor for all flavoenzymes tested. 

Mechanistically, gold(I) compounds can undergo a ligand exchange reaction with the thiolates and selenolates in the flavoenzyme’s catalytic site, resulting in enzyme–gold adducts [35,36]. To further understand the inhibition of Cys-TrxRs by hybrid compounds, we set up a cell-free assay to monitor the consummation of reduced glutathione (GSH) as a mimetic model for nucleophiles. Initially, GSH was incubated for 2 or 24 h at compound concentrations ranging from 10 to 0.3 drug:GSH molar ratio, and the consummation of GSH was determined by fluorescence of monochlorobimane (Appendix A). When compounds were incubated for 2 h, [AuClPPh_3_] was able to interact with GSH in a drug-concentration manner, while neither CQ nor phenanthroline interacted with GSH (negative controls). [AuClPPh_3_] was more potent than [AuAQPQ]PF_6_ (**2**) in interacting with GSH; however, when compounds were incubated for 24 h, binding of [AuClPPh_3_] for GSH remained almost unaltered, and presumably, it reacted with GSH by a ligand exchange reaction. Differently, the binding of [AuAQPQ]PF_6_ (**2**) with GSH increased during 2 versus 24 h incubation, indicating that, over time, [AuAQPQ]PF_6_ (**2**) was consuming GSH. AQ interaction with GSH had a similar behavior: its reaction after 2 h incubation seems to be incomplete; meanwhile, at 24 h incubation, GSH consummation increased. It was interpreted that the interaction of [AuAQPQ]PF_6_ (**2**) with GSH is more similar to that observed for AQ than for [AuClPPh_3_]. 

To reproduce the findings above and to characterize them at the molecular level, the interaction of compounds with GSH was monitored by NMR. After ascertaining the interaction of quinolinic compounds PQ and AQ with GSH, the GSH interaction with [AuAQPQ]PF_6_ was analyzed (Appendix A). This step allowed us to unequivocally ascertain that GSH binds to the quinolinic ring of the primaquine component of hybrid (**2**). GSH conjugation to the primaquine component of hybrid (**2**) was inferred by the absence of the peak corresponding to Hg hydrogen of PQ after the addition of GSH. This behavior was also observed in an experiment using metal-free PQ (data not shown), and it was previously observed by others [37]. Gold(I) reduction to Au(0) species, ligand-exchange reactions or the appearance of the signals corresponding to metal-free quinolinic ligands were not observed. Although both PQ and AQ can bind to GSH, because of the dissimilar mode of coordination, the PQ component in the complex is more prone to bind to GSH than to gold(I) or AQ.

## 4. Discussion

We have demonstrated the chemistry, potency and efficacy of [AuAQPQ]PF_6_ (**2**) as an antimalarial agent. In the strategy to synthesize [AuAQPQ]PF_6_ (**2**), the PQ component was coordinated with the gold(I) atom via its aliphatic nitrogen, while the AQ component was coordinated via its quinolinic nitrogen. This mode of coordination for AQ was an unexpected finding, especially considering that in the same synthesis strategy for its analog [AuCQPQ]PF_6_ (**1**), the CQ component was coordinated with gold(I) through its aliphatic nitrogen. Despite the dissimilarity in terms of metal coordination, [AuAQPQ]PF_6_ (**2**) was quite stable in the air, in aqueous solution, and in DMSO solutions as previously observed for its analog (**1**).

Against the asexual blood stages of *P. falciparum*, [AuAQPQ]PF_6_ (**2**) was twice less potent than AQ for the drug-sensitive strain but was significantly more potent than AQ for the drug-resistant strain. This is because in the sensitive strain, the activity of [AuAQPQ]PF_6_ (**2**) is largely due to the presence of the AQ component, while PQ makes little or no contribution to antimalarial activity in this stage. However, PQ can contribute to antimalarial activity against drug-resistant parasite strains since these strains are typically hypersensitive to treatment [16,21]. Moreover, toxicity to mammalian cells was low for the hybrid (**2**), resulting in a high selectivity index. 

The efficacy of [AuAQPQ]PF_6_ (**2**) against the asexual blood stages using *P. berghei*-infected mice revealed that it is a powerful antimalarial agent. It was at least three times more efficacious than a combination of its components (AQ + PQ). PQ has low potency and subsequently limited efficacy against asexual blood stages. Conceivably, the summation of the efficacies against these stages upon PQ co-administration could only be achieved if a high PQ dosage was deployed. However, [AuAQPQ]PF_6_ (**2**) is composed of a fixed ratio of AQ + PQ; therefore, the superior efficacy of [AuAQPQ]PF_6_ (**2**) versus a drug combination of AQ + PQ could be partly due to its pharmacokinetic profile and an increase in the cellular accumulation of PQ in the parasite cells upon [AuAQPQ]PF_6_ (**2**) exposure. While further investigation of this interpretation is beyond the scope of the present study, we note that previous studies have also observed a potency enhancement of *in vivo* efficacy for hybrid molecules containing PQ in comparison to PQ alone or its drug combination [38,39]. 

[AuAQPQ]PF_6_ (**2**) has an improved spectrum of inhibitory activity against multiple stages of the *Plasmodium* life cycle, spanning from sporozoites to gametocytes. The enhanced potency of the hybrid compared to drug combinations is most likely due to a cooperation of two quinolinic drugs with distinct modes of actions; specifically, PQ, causing an augmentation of oxidative species toxic for the parasite [20], and AQ, suppressing heme detoxification and decreasing the levels of antioxidant GSH in egressing hepatic schizonts and in young gametocytes [19,40]. The reason for the superior potency of [AuAQPQ]PF_6_ over a drug combination remains less clear, but it is plausible that it affects thiol redox homeostasis while metal-free quinolines do not. 

Mechanistically, the activity of [AuAQPQ]PF_6_ (**2**) could be partly ascribed to the suppression of heme detoxification, inhibition of the activity of flavoenzymes involved in the thiol redox homeostasis, and the cooperation of both effects. In support of this hypothesis, [AuAQPQ]PF_6_ (**2**) binds to Fe(III)-PPIX and inhibits β-hematin crystal formation, demonstrating the potential of (**2**) to suppress parasite heme detoxification. In contrast, [AuAQPQ]PF_6_ (**2**) inhibits flavoenzyme activity, including *Pf*TrXR, relatively weakly in comparison to its inhibition of in vitro parasite growth. We observed that while both [AuAQPQ]PF_6_ (**2**) and its quinolinic components react with GSH, they do so in a different way from [AuCl(PPh_3_)]: while the latter reacts quickly with GSH, the former react slowly. Furthermore, the ^1^H-NMR analysis did not display evidence of ligand exchange, such as a displacement of quinoline from the sphere of coordination of gold(I), but it did display evidence of GSH binding to the PQ component of [AuAQPQ]PF_6_ (**2**). Therefore, the multiple analyses suggested that [AuAQPQ]PF_6_ (**2**) has relatively low reactivity for thiol reactants, indicating that quinoline dissociation in the hybrid compounds may only occur inside parasite cells. Moreover, this may explain the relatively low inhibitory activity of [AuAQPQ]PF_6_ (**2**) for recombinant flavoenzymes. 

The dissimilarity in the coordination of 4-aminoquinoline CQ or AQ to the gold(I) atom has implications for the mechanism of action of hybrid compounds as well as for their antimalarial activity. Our previous hybrid compound [AuCQPQ]PF_6_ (**1**) was more potent in inhibiting β-hematin than CQ, while [AuAQPQ]PF_6_ (**2**) was equipotent to AQ. It is well known that CQ and AQ interact with Fe(III)-PPIX by iron–quinoline coordination and the stacking of quinoline over protoporphyrin [41,42]. In the [AuCQPQ]PF_6_ (**1**) hybrid, the 4-aminoquinolinic heterocyclic is available to interact with Fe(III)-PPIX by both modes, while in the [AuAQPQ]PF_6_ (**2**) hybrid, quinoline is coordinated with gold(I). Thus, only stacking interactions between Fe(III)-PPIX and (**1**) are possible. This is one explanation for why [AuAQPQ]PF_6_ (**2**) did not display superior potency to AQ as a heme detoxification suppressor (Figure 4). The coordination of 4-aminoquinoline CQ or AQ to gold(I) also has consequences for the mechanisms affecting thiol redox homeostasis, and it can be inferred by observing that [AuCQPQ]PF_6_ (**1**) is more potent in inhibiting the enzymatic activity of flavoenzymes than [AuAQPQ]PF_6_ (**2**). This supports the hypothesis that compound (**1**) is more prone to interact with flavoenzymes than compound (**2**); namely, [AuCQPQ]PF_6_ (**1**) is liable to undergo a ligand exchange reaction [43,44] in the presence of flavoenzymes. While much work remains to be conducted to clearly define the gold(I) hybrids’ effect on thiol redox homeostasis in parasite cells, we were intrigued to find that the metallic hybrids displayed selectivity for inhibiting Sec-TrxRs versus Cys-TrxR, indicating that the gold(I) compounds share some reactivity to interact with Sec-TrxR. This agrees with the prevailing model that the inhibitory activity of gold(I) compounds for flavoenzymes is explained by the kinetics of a ligand exchange reaction rather than the selective affinity of gold species for Sec versus Cys [35]. 

## 5. Conclusions and Perspectives

The [AuAQPQ]PF_6_ (**2**) compound is of remarkable stability, of facile preparation and displays high potency and strong efficacy as a blood schizonticidal agent and of improved spectrum of activity against multiple stages of the *Plasmodium* life cycle. It achieves suppression of heme detoxification process, as inferred by drug binding to [Fe(III)PPIX] and β-hematin inhibitory activity (BHIA), and it may cause an imbalance in the thiol redox homeostasis of the parasite, as inferred by its inhibition on the enzymatic activity of flavoenzymes. Future work needs to determine the contribution of each mechanism (heme detoxification versus redox homeostasis) for hybrid antiparasitic activity, but the findings presented here indicate the mutual benefits of combining gold(I) to quinolines. Thus, this study provides compelling evidence that hybrid-based drugs featuring gold(I) as a chemical linker represent a potential approach for drug design. 

## Figures and Tables

**Figure 1 pharmaceutics-14-01251-f001:**
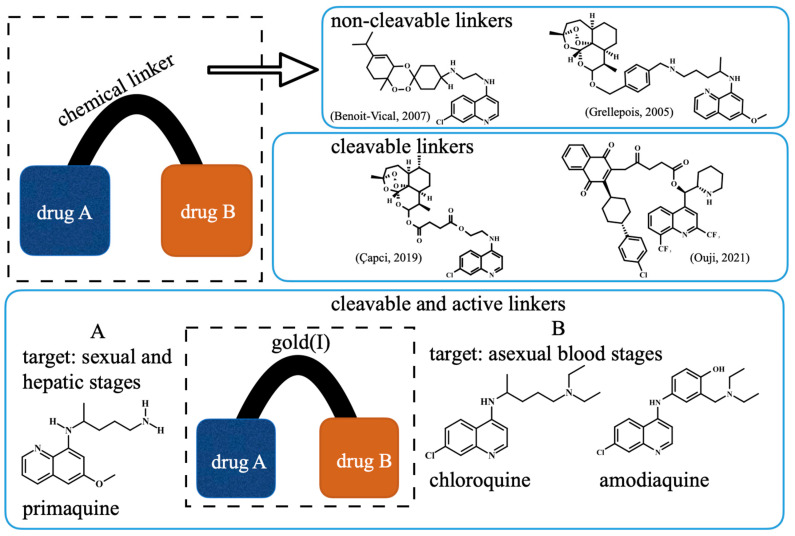
Panel (**A**) Schematic of the concept of hybrid-based drugs against malaria, highlighting the nature of chemical linkers and showing representative hybrids containing aminoquinolines as one of the components (references given in brackets). Panel (**B**) The concept of designing hybrid compounds containing gold(I) as a dual cleavable chemical linker and an antimalarial payload [9,10,11,12].

**Figure 2 pharmaceutics-14-01251-f002:**
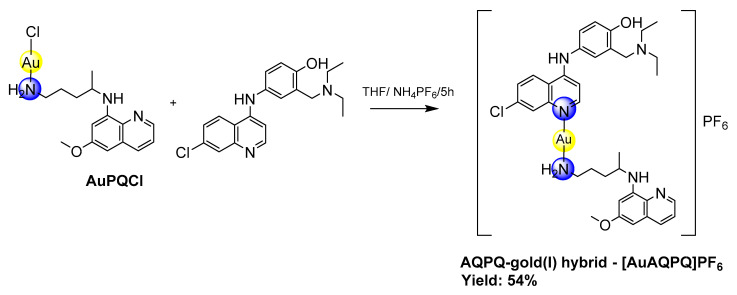
Synthesis of [AuAQPQ]PF_6_ (**2**).

**Figure 3 pharmaceutics-14-01251-f003:**
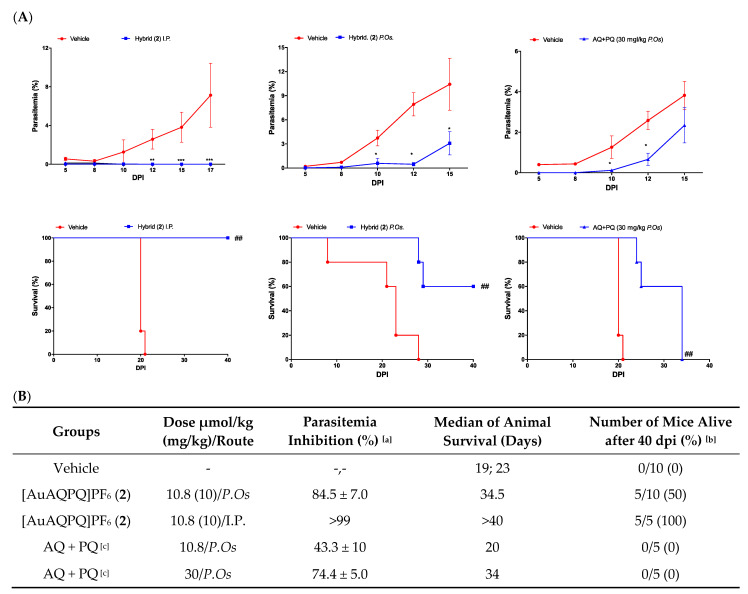
Efficacy of compounds to suppress the parasitemia in *P. berghei*-infected Swiss mice. Panel (**A**): Representative graphs of parasitemia and animal survival. Panel (**B**): Table summarizing the results. Footnotes for table: ^[a,b]^ Values are from a single experiment, using *n* = 5/group unless indicated. ^[a]^ Values are mean and standard deviation determined in comparison to vehicle group. ^[c]^ Refers to a 1:1 mixture of AQ and PQ. * *p* < 0.05, ** *p* < 0.01, *** *p* < 0.001 (one-way ANOVA) versus vehicle. ^##^
*p* < 0.05 (log-rank and Mantel–Cox test) versus vehicle. DPI, days post-infection. *P.Os.,* orally by gavage; I.P., intraperitoneal injection.

**Figure 4 pharmaceutics-14-01251-f004:**
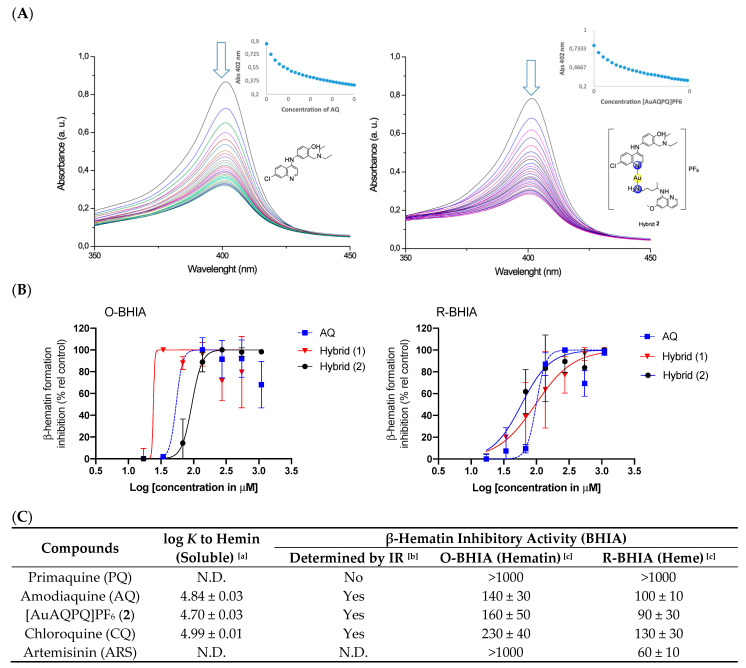
Effects of compounds on the suppression of heme detoxification as inferred by drug:hemin binding and inhibition of the β-hematin formation. Panel (**A**) Titration of ferriprotoporphyrin IX [Fe(III)-PPIX, hemin]. Panel (**B**) Inhibition of the β-hematin formation; dots are the median ± S.E.M and lines are the data fitted into the log(concentration) versus response with a variable slope equation. Panel (**C**) Table summarizing the properties. Arrow in panel (**A**) indicates the decrease in absorbance upon increasing concentration of compounds. Footnotes for table: ^[a]^ Association constant to [Fe(III)-PPIX]. Values are median ± S.E.M. of three independent experiments. ^[b]^ β-hematin formation upon incubation of a 3:1 ratio of drug:hematin for 30 min. and qualitatively determined by infrared (IR). Yes indicates β-hematin inhibition; No indicates β-hematin formation. ^[c]^ β-hematin formation determined after 18 h of incubation. Values are median ± S.E.M. of IC_50_ μM from three independent experiments. O-BHIA, oxidizing BHIA using hematin as reactant; R-BHIA, reducing BHIA using heme as reactant; S.E.M., standard error of the mean; AQ, amodiaquine (free base); PQ, primaquine (free base); CQ, chloroquine (free base); ARS, artemisinin; N.D., not determined.

**Figure 5 pharmaceutics-14-01251-f005:**
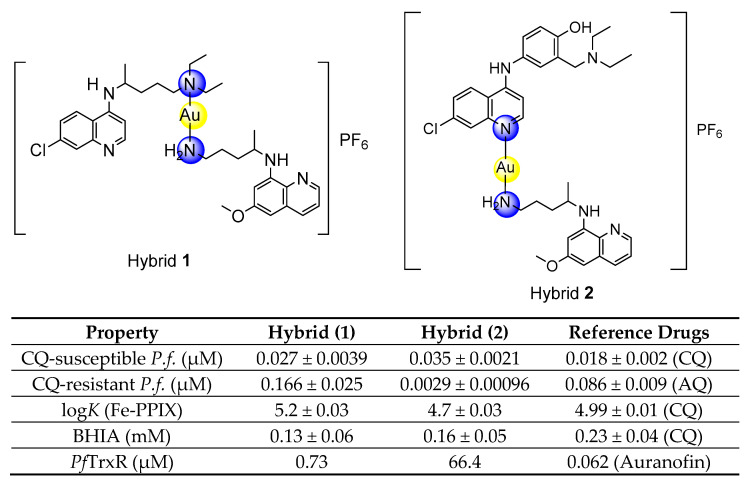
Summary of the structure–activity relationships and drug properties for the gold(I) hybrids. CQ, chloroquine; AQ, amodiaquine.

**Table 1 pharmaceutics-14-01251-t001:** In vitro activity of compounds against the asexual blood stages of *P. falciparum*, cytotoxicity for mammalian cells and the selectivity profile.

Compounds	*P. falciparum*, IC_50_ ± S.E.M. (nM) ^[a]^		CC_50_ ± S.E.M. (nM) ^[b]^	Selectivity Index ^[c]^	
	CQ-Susceptible 3D7	CQ-Resistant W2		3D7	W2
Primaquine (PQ)	1117 ± 53	182 ± 16	36,500 ± 1100	32	200
Amodiaquine (AQ)	18.0 ± 1.6	86.2 ± 9.4	23,800 ± 700	1322	276
[AuAQPQ]PF_6_ (**2**)	35.5 ± 2.1	2.97 ± 0.96	16,400 ± 700	461	5655
Chloroquine (CQ)	18.0 ± 2.0	459 ± 21	51,500 ± 1860	2861	112

^[a]^ Inhibitory concentration for 50% (IC_50_) determined 72 h after incubation with compounds using the histidine-rich protein 2 ELISA kit. Values were calculated as mean ± S.E.M. of at least three independent experiments. ^[b]^ Cytotoxic concentration for 50% (CC_50_) against the macrophages of the J774 cell lineage, determined 72 h after incubation with compounds using the CellTiterGlo kit. Values were calculated as mean ± S.E.M. of at least two independent experiments. ^[c]^ Determined as CC_50_/IC_50_. S.E.M., standard error of the mean; AQ, amodiaquine hydrochloride hydrate; PQ, primaquine diphosphate; CQ, chloroquine diphosphate.

**Table 2 pharmaceutics-14-01251-t002:**
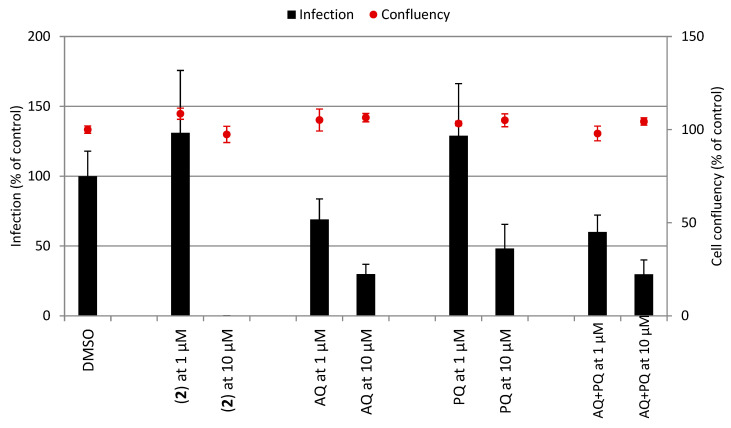
In vitro activity of compounds against multiple stages of *Plasmodium* life cycle. Inset shows the infection of Huh-7 cells by *P. berghei* sporozoites (black bars) and the Huh-7 cell confluency (red dots).

Compounds	IC_50_ (µM) and Parasite Stages			
	Sporozoites (Inhibition of Infection, %) ^[a]^	Uninfected Hepatic Cells ^[b]^	GAMETOCYTES of *P. falciparum* ^[c]^	
			Stages I/III	Stage V
Primaquine (PQ)	52	>80	13.7 ± 7.6	43.2 ± 10.6
Amodiaquine (AQ)	71	>80	0.061 ± 0.02	23.6 ± 5.0
[AuAQPQ]PF_6_ (**2**)	100	14.4 ± 2.7	0.035 ± 0.01	9.5 ± 3.1
AQ + PQ ^[d]^	70	41.0 ± 3.2	N.D.	N.D.
Reference	52 (PQ)	<0.12 (DOXO)	0.0123 ± 0.0054 (DHA)	0.038 ± 0.014 (MB)

^[a]^ Assay against Huh-7 cells infected by *P. berghei* sporozoites and activity was determined after 48 h of drug (10 µM) incubation. Values are the % of inhibition in comparison to untreated control from one experiment. ^[b]^ Cytotoxicity in uninfected HepG2 cells determined 72 h after drug incubation. ^[c]^ Assay against gametocytes of 3D7 strain of *P. falciparum* and activity was determined after 72 h. Values are mean ± S.D. of one experiment, with each concentration tested in triplicate. ^[d]^ Refers to 1:1 mixture of AQ and PQ. Reference drug for each assay: DOXO, doxorubicin; DHA, dihydroartemisinin; MB, methylene blue.

**Table 3 pharmaceutics-14-01251-t003:**
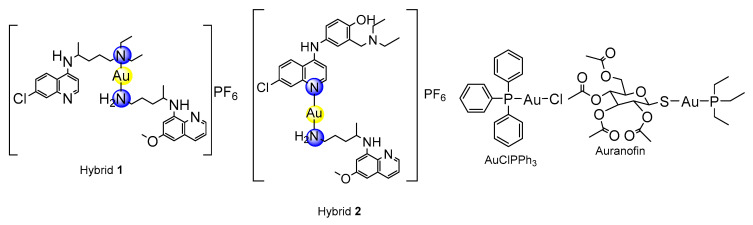
Effects of compounds on the thiol redox homeostasis as inferred by the inhibitory effects on the enzymatic activity of recombinant flavoproteins.

Compounds	IC_50_ (μM) for TrxRs ^[a]^			Human GR ^[a]^	*P. falciparum* 3D7 (Cell Based Activity) ^[c]^
	Human (Sec)	*S. mansoni* (Sec) ^[b]^	*P. falciparum* (Cys)		
Amodiaquine (AQ)	26.5	42.0	>66.7	>66.7	0.018 ± 0.002
[AuAQPQ]PF_6_ (**2**)	4.98	0.75	66.4	>66.7	0.035 ± 0.0021
[AuCQPQ]PF_6_ (**1**)	0.0068	0.0038	0.73	0.053	0.027 ± 0.0039
[AuClPPh_3_]	0.00065	0.0005	0.0019	0.00118	4000 ± 512
Auranofin	0.020	0.007	0.062	40.0	0.060 ^[d]^

^[a]^ IC_50_ values are in μM and the mean of one experiment, where each drug was tested in five different concentrations in triplicate. TrxR, thioredoxin reductase; GR, glutathione reductase. ^[b]^
*S. mansoni* thioredoxin–glutathione reductase (*Sm*TGR). **^[c]^** In vitro antiplasmodial activity expressed as IC_50_ values in μM. ^[d]^ Value taken from reference [25]. Sec, redox-active site composed of selenocysteine; Cys, redox-active site composed of cysteine.

## Data Availability

Supporting information is available enclosed.

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
