# Peer review of "A Hybrid of Amodiaquine and Primaquine Linked by Gold(I) Is a Multistage Antimalarial Agent Targeting Heme Detoxification and Thiol Redox Homeostasis"

_pharmaceutics, 2022, doi:10.3390/pharmaceutics14061251_

Round 1
Reviewer 1 Report
The manuscript, "Hybrid of amodiaquine and primaquine linked by gold(I) is a multistage antimalarial agent by targeting heme detoxification and thiol redox homeostasis," by Pereira, et al. is, in the opinion of this reviewer, interesting and worthy of publication once some issues are addressed.
First, there are issues with language throughout, although not severely so. But the title might better be, "A hybrid of amodiaquine and primaquine linked by gold(I) is a multistage antimalarial agent by targeting heme detoxification and thiol redox homeostasis". Normally, this reviewer would let this go, but setting the title correctly would be important.
Then, line 74 starts with the word, combo. This is a very informal abbreviation, and should be combination. This reviewer was enjoying reading the manuscript at this point, but found this 'word' to be jarring.
Check spelling throughout. For example, synthetized - in line 115.
The synthesis of the topic compound (2): Purity was never really addressed. This is really problematic because the integrals in the 1H NMR spectra (Figures S2 and S3) really do not point to a 1:1 ratio of the two halves of the hybrid. Here this reviewer sees 0.8:1.1:0.7 (for example), instead of 1.0:1.0:1.0. The spectrum could be re-collected with a substantially longer relaxation delay (e.g., 10 s), and perhaps using the Bruker pulse program zg30. This might clear this up, but the baseline in the aromatic portion (6 - 9 ppm) is not really 'clean', and so this reviewer really is worried about purity of Compound (2). Also, no HPLC analysis is reported (same concern: purity).
Also related to the NMR spectroscopy, arguments are made, beginning at the bottom of Page 9, regarding the coordinating ligand identities (aliphatic & quinoline Ns) using chemical shift arguments, which this reviewer finds not quite convincing. Because the compound is obtained as a crystal, might it be possible to get crystallographic evidence as to the actual structure? Alternatively, 2D NOESY might find spatial proximity between the PQ NH2 (or the adjacent CH2) and the AQ H-2 and/or H-8? Such evidence would be direct and more convincing than the chemical shift arguments presented. This reviewer does consider the 13C chemical shift argument to be more convincing, although this could have been made more clear by showing a simple 1H-13C HSQC (or HMQC) spectrum that would give the simple correlations between the directly-bonded 1H and the corresponding 13C resonances. This reviewer questions how much the typical reader will get from the HMBC spectra, compared to the value of one 1H-13C HSQC plot.
The structures of PQ and AQ in Figure S3 appear to be switched, with respect to the appropriate spectra.
The stability, measured for DMSO-d6 solution (Figure S11) does not seem to be very relevant. Why was this not performed in D2O (or better, 90% H2O/10% D2O)?
The spectra in the Supplementary section should contain more information, so as to be immediately available to the reader upon examining the data. This information should include at least temperature, field strength (MHz), and solvent.
The functional assays presented through the rest of the manuscript are generally fine, as presented. However the Discussion makes stronger claims than this reviewer can support. In particular, the Discussion begins with the statement, "We have demonstrated the chemistry, potency, efficacy, and mechanism of action of [AuAQPQ]PF6 (2) as an antimalarial agent." In fact, the authors have presented evidence of chemistry, potency, etc.
Author Response
Reviewer 1:
First, there are issues with language throughout, although not severely so. But the title might better be, "A hybrid of amodiaquine and primaquine linked by gold(I) is a multistage antimalarial agent by targeting heme detoxification and thiol redox homeostasis". Normally, this reviewer would let this go, but setting the title correctly would be important. Then, line 74 starts with the word, combo. This is a very informal abbreviation, and should be combination. This reviewer was enjoying reading the manuscript at this point, but found this 'word' to be jarring. Check spelling throughout. For example, synthetized - in line 115.
Response: We thank this reviewer for the careful review, for the important suggestions to improve the chemical part of the manuscript and for appreciating the interest and quality of our work. The title of manuscript was modified accordingly to the above suggestion.
Reviewer 1: The synthesis of the topic compound (2): Purity was never really addressed. This is really problematic because the integrals in the 1H NMR spectra (Figures S2 and S3) really do not point to a 1:1 ratio of the two halves of the hybrid. Here this reviewer sees 0.8:1.1:0.7 (for example), instead of 1.0:1.0:1.0. The spectrum could be re-collected with a substantially longer relaxation delay (e.g., 10 s), and perhaps using the Bruker pulse program zg30. This might clear this up, but the baseline in the aromatic portion (6 - 9 ppm) is not really 'clean', and so this reviewer really is worried about purity of Compound (2). Also, no HPLC analysis is reported (same concern: purity).
Response:
We agree with the reviewer that the purity of the final compound is critical for this study, especially because it involves pharmacological evaluation. The synthesis of compound (2) was performed using quinolines, which are free-base instead of commercially available quinoline salts. While this requires extensive purification steps and is much more labourious, it allows for the washing of the resulting crude material with organic solvents in order to remove unreacted quinolines.
The purity of the final compound was determined by combustion analysis for C,H,N. This analysis was repeated several times, as several batches were needed for in vivo evaluation in mice. The values obtained for C,H,N of compound 2 are in agreement to the calculated values, therefore, we are confident that the final compound has adequate purity.
We agree with the utilization of liquid chromatography recommended by the reviewer; however, we currently do not have available protocol for determining purity through a chromatography analysis. Since the purity of final compound by combustion is in very good standard, we feel quite confident about the reproductibility of the synthesis and purity of compounds.
Finally, we agree with the reviewer’s comment that “some but not all integrals in the resonance do not match to an expected 1:1:1 ratio”. Here, we did not vary the time of relaxation because it was not the case of a missing signal but because of integration of signals. We have revised the integrals in the current version, where most integrals matched the ratio. Instances where intergrals do not match ratio is a result of the delay in the time of relaxation and not impurity. We have observed this same phenomena for other metal complexes containing quinolines (see Inorg Chem. 2015 Dec 21;54(24):11709-20, DOI 10.1021/acs.inorgchem.5b01647; J Inorg Biochem. 2015 Dec;153:150-161. doi: 10.1016/j.jinorgbio.2015.07.016).
Based on these explanations, we hope that the reviewer agrees that the chemical characterization and purity is correct and satisfactory for the final compound.
Also related to the NMR spectroscopy, arguments are made, beginning at the bottom of Page 9, regarding the coordinating ligand identities (aliphatic & quinoline Ns) using chemical shift arguments, which this reviewer finds not quite convincing. Because the compound is obtained as a crystal, might it be possible to get crystallographic evidence as to the actual structure? Alternatively, 2D NOESY might find spatial proximity between the PQ NH2 (or the adjacent CH2) and the AQ H-2 and/or H-8? Such evidence would be direct and more convincing than the chemical shift arguments presented. This reviewer does consider the 13C chemical shift argument to be more convincing, although this could have been made more clear by showing a simple 1H-13C HSQC (or HMQC) spectrum that would give the simple correlations between the directly-bonded 1H and the corresponding 13C resonances. This reviewer questions how much the typical reader will get from the HMBC spectra, compared to the value of one 1H-13C HSQC plot.
Response: First, we would like to note that although final compound was obtained as a crystal, the quinolines amodiaquine and chloroquine are difficult to precipate into crystals suitable for X-ray diffraction; there is a paucity of X-ray studies providing metal complexes with these ligands. Most X-ray studies of metal complexes are indeed for modified quinolines as ligands (see Inorg Chem. 2015 Dec 21;54(24):11709-20, DOI 10.1021/acs.inorgchem.5b01647; J Inorg Biochem. 2015 Dec;153:150-161. doi: 10.1016/j.jinorgbio.2015.07.016).
Second, we agree with the reviewer that a correct determination of coordination sphere of the final compound is critical for this study. From the reviewer’s comments we feel that we did not explain our spectroscopical methods with sufficient clarity in the first version of the manuscript. Therefore, we have now recorded 13C-1H HMBC spectrum and added it to Figure S6 (supporting material) which provides compelling support for the chemical attribution of 13C signals.
Regarding the possibility of recording a 2D NOESY analysis, while this is a very genuine suggestion, we posit it would not show any additional information because the signal corresponding to NH2 does not appear in the 1HNMR spectrum due to its exchange with D2O, thus, it would not be possible to see the spatial proximity correlation between the PQ NH2 and the AQ H-2 and/or H-8.
The structures of PQ and AQ in Figure S3 appear to be switched, with respect to the appropriate spectra.
Response: We are sorry for this mistake. It has been corrected in the current version.
The stability, measured for DMSO-d6 solution (Figure S11) does not seem to be very relevant. Why was this not performed in D2O (or better, 90% H2O/10% D2O)?
Response: From the reviewer’s comments we feel that we did not explain the relevance of using DMSO with sufficient clarity in the first version of the manuscript.
We would like to clarify that we studied the stability of compound 2 in DMSO because it was used to dissolve this compound for the pharmacological assays. Knowing that DMSO is a good coordinating ligand, it was important to see if there any sort of ligand exchange reaction that can take place. Here, we demonstrated that the gold hybdrid is stable in DMSO in these conditions. It was our plan to study the stability of the compound in an aqueous medium by a combination of UV-Vis and NMR analyses, but due to solubility issues we could not perform these experiments in time. Of note, we observed that during the experiments in the presence of GSH, which was performed in a mixed DMSO-D2O medium, there was no apparent ligand exchange reaction of the compound, indicating a certain degree of stability in an aqueous medium.
The spectra in the Supplementary section should contain more information, so as to be immediately available to the reader upon examining the data. This information should include at least temperature, field strength (MHz), and solvent.
Response: We are sorry these data were missing. It has been added in the current version.
The functional assays presented through the rest of the manuscript are generally fine, as presented. However the Discussion makes stronger claims than this reviewer can support. In particular, the Discussion begins with the statement, "We have demonstrated the chemistry, potency, efficacy, and mechanism of action of [AuAQPQ]PF6 (2) as an antimalarial agent." In fact, the authors have presented evidence of chemistry, potency, etc.
Response: This sentence has been revised in the current version.
Reviewer 2 Report
This is a wonderful paper with respect to the content and its presentation, to the scientific concept and realization and the possible pharmaceutical consequences in Malaria therapy. It might be appropriate now to repeat the relevant ideas and results behind this paper, but the Abstract does this already in a highly convincing way. Just mention the abbreviations AQ and PQ already in line 4 of this Abstract. I went through the paper two times, and to my regret, I could not find any bad spots in the chemistry, spectroscopy, biological analysis and the overall interpretation of the results. Especially the improved activity against drug-resistant P. falcuparum strains is highly relevant and deserves further optimization and hybrid modification. It stimulates all groups active in following the hybrid concept in drug development. This paper can be published as is.
Author Response
Response: We thank the reviewer for the kind words and for the recognizion of the quality of our manuscript. Abbreviations for amodiaquine and primaquine were included in the current version of manuscript.
Round 2
Reviewer 1 Report
This revised manuscript is fine for publication, in the opinion of this reviewer.